# Flexible Kinetic Model Determination of Reactions in Materials under Isothermal Conditions

**DOI:** 10.3390/ma16051851

**Published:** 2023-02-23

**Authors:** Juan Arcenegui-Troya, Antonio Perejón, Pedro E. Sánchez-Jiménez, Luis A. Pérez-Maqueda

**Affiliations:** 1Instituto de Ciencia de Materiales de Sevilla, Consejo Superior de Investigaciones Científicas-Universidad de Sevilla, C. Américo Vespucio no 49, 41092 Sevilla, Spain; 2Departamento de Química Inorgánica, Facultad de Química, Universidad de Sevilla, 41012 Sevilla, Spain

**Keywords:** kinetic analysis, kinetic modelling, model free, polymer degradation, isothermal

## Abstract

Kinetic analysis remains a powerful tool for studying a large variety of reactions, which lies at the core of material science and industry. It aims at obtaining the kinetic parameters and model that best describe a given process and using that information to make reliable predictions in a wide range of conditions. Nonetheless, kinetic analysis often relies on mathematical models derived assuming ideal conditions that are not necessarily met in real processes. The existence of nonideal conditions causes large modifications to the functional form of kinetic models. Therefore, in many cases, experimental data hardly obey any of these ideal models. In this work, we present a novel method for the analysis of integral data obtained under isothermal conditions without any type of assumption about the kinetic model. The method is valid both for processes that follow and for those that do not follow ideal kinetic models. It consists of using a general kinetic equation to find the functional form of the kinetic model via numerical integration and optimization. The procedure has been tested both with simulated data affected by nonuniform particle size and experimental data corresponding to the pyrolysis of ethylene-propylene-diene.

## 1. Introduction

The philosophy of kinetic analysis in materials is to determine the parameters and the physico-geometrical model that describe gas–solid or solid–solid reactions to make predictions about their progress in a wide range of conditions. It is a powerful tool with applicability to a myriad of processes that lie at the core of the materials manufacturing industry and materials applications [1,2,3,4]. Isothermal kinetic analysis (IKA) is extensively employed for studying a large number of heterogeneous processes, including calcium looping under realistic conditions [5,6], curing [7,8,9], hydration and dehydration [10,11], redox [12,13], and crystallization [14]. From the experimental point of view, isothermal experiments are straightforward as temperature is maintained constant while the evolution of the process is monitored by changes in mass, heat flow, spectroscopic or diffraction measurements, changes in volume or dimensions, changes in physical properties, etc. [15]. Thus, IKA is used to analyse data recorded with a large variety of techniques, including thermogravimetry [16,17], differential scanning calorimetry [18], in situ XRD synchrotron [19,20], and Raman spectroscopy [21]. Moreover, from the kinetic point of view, a significant advantage of IKA is related to the fact that the shape of the extent of the reaction curve is directly linked to the kinetic model [22], unlike linear heating rate experiments where the curves always exhibit a sigmoidal shape regardless of the obeyed model [23,24,25]. Moreover, IKA simplifies the determination of the apparent activation energy, as it has been demonstrated that the resulting values of activation energy are independent of the assumed kinetic model [26,27].

The model-fitting approach is, by far, the most common procedure for IKA. Thus, the model from the list of literature ideal kinetic models that provides the best fit is considered to be the true kinetic model of the process [28,29]. Thus, it is assumed that the process follows one of these ideal models. Moreover, although, as stated before, the value of the calculated activation energy is independent of the assumed model, it is crucial to perform accurate predictions on the reaction’s evolution with time [30]. A problem arises when none of the ideal models can properly describe the process. Actually, models are deduced assuming ideal conditions that are not necessarily met in real systems. For instance, it is known that the particle size distribution (PSD) strongly affects the kinetics and leads to a significant deviation from ideal models, which are deduced assuming that all particles are identical [31,32,33]. Consequently, ideal models are often insufficient to adequately describe the reaction’s progress.

Perez-Maqueda et al. proposed using an empirical equation, a modified version of the Setak–Berggren equation (MSBE), that covers most kinetic models found in the literature and their deviation due to nonideal conditions present in experiments [34]. These authors applied the MSBE to combine kinetic analysis, which simultaneously analysed differential data recorded under different conditions. Unfortunately, the integral of the MSBE does not have an analytical solution, and therefore, MSBE could not be directly applied to the analysis of integral isothermal data.

Here, we propose a novel method that uses the MSBE to calculate, through the optimization and numerical integration of data recorded under isothermal conditions, the kinetic model obeyed by the reaction without any assumption about it. Using numerical simulations, we prove that the method can be effectively applied to reactions described by models modified by nonideal conditions. Finally, the procedure is also tested with its application to the kinetic analysis of experimental data (the thermal decomposition of ethylene-propylene-diene).

## 2. Experimental Section

The ethylene-propylene-diene (EPDM) used in this work was supplied by ExxonMobil (Vistalon Rubbers, Barcelona, Spain): 69 wt% ethylene, 5 wt% ethylidene norbornene, and 26 wt% propylene. The thermal degradation of EPDM was studied under isothermal conditions using a thermogravimetric analyser Q5000IR from TA instruments (New Castle, DE, USA). The pyrolysis was conducted at 683, 693, 698, and 703 K. The experiments started with a heating ramp at 300 K·min^−1^ from room temperature to the temperature of the isotherm. Then, the temperature was maintained constant for two hours in the experiments with EPDM. The experiments were carried out in nitrogen with a flow rate of 200 mL/min. Since methods for kinetic analysis are quite sensible to errors associated with measurements, the influence of experimental conditions needed to be carefully considered when recording data [35,36]. For instance, the conclusion of kinetic analysis can be strongly conditioned by the flow rate used to conduct the experiments or/and the characteristics of the reactor employed [37]. The EPDM thermal degradation experiments were conducted with small amounts of material (10 mg) to prevent heat and mass transfer phenomena and obtain experimental data representative of the reaction.

## 3. Theoretical Foundation and Simulations

In general, the progress of solid–gas reactions far from equilibrium can be described by [38]:(1)$$\frac{d\alpha }{dt}=A\cdot \exp\left(-\frac{E}{RT}\right)\cdot f\left(\alpha \right)$$
where A is the pre-exponential factor, E is the apparent activation energy, T stands for temperature, t is the time, α is the extent of reaction (calculated as the ratio between the mass converted at the instant t to the total mass converted once the process is completed), and f(α) is the kinetic model that describes the progress of the process. Under isothermal conditions (T=constant), Equation (1) becomes:(2)$$\frac{d\alpha }{dt}=k\cdot f\left(\alpha \right)$$

k is the reaction rate constant. This later equation can be integrated to obtain a relationship between α and t:(3)g(α)=k·t
where:(4)$$g\left(\alpha \right)=\int \frac{d\alpha }{f\left(\alpha \right)}$$
is the integral form of the kinetic model. Using a modified Sestak–Bergen equation (MSBE) originally proposed in [34] as an empirical kinetic model:(5)$$f\left(\alpha \right)=C{\left(1-\alpha \right)}^{n}\alpha^{m}$$
where *C*, *n,* and *m* are the parameters to be determined from the best fitting to experimental data. Equation (4) can be written as:(6)$$g\left(\alpha \right)=\int \frac{d\alpha }{{\left(1-\alpha \right)}^{n}\alpha^{m}}$$

Since experimental data are discretely recorded, we need to approximate Equation (6) with a sum. Then, the integral form of g at the instant $t=t_{N}$ is estimated by:(7)$$g\left(t=t_{N},n,m\right)=\overset{N}{\underset{i=0}{\sum }}\frac{\alpha \left(t_{i+1}\right)-\alpha \left(t_{i}\right)}{{\left(1-\alpha \left(t_{i}\right)\right)}^{n}\alpha {\left(t_{i}\right)}^{m}}$$
where N is the number of measurements up to the instant $t=t_{N},$ and $\alpha \left(t_{i}\right)$ is the extent of the reaction at $t=t_{i}$. The lapse of time between two consecutive measurements is $\Delta t=t_{i+1}-t_{i}$ with $t_{0}=0$ s. With this approximation, Equation (3) becomes:(8)$$g\left(t=t_{N},n,m\right)=kt_{N}$$

The linear relationship will be only satisfied for a particular selection of the values for n and m corresponding to the empirical kinetic model expressed by Equation (5). Hence, it is an optimization problem that can be algorithmically solved by trying different combinations of n and m while assessing the R-squared correlation coefficient ($R^{2}$) when representing g versus t to see which combination yields the closest value to 1. In this work, we employed a method based on the approximation to the optimum value. We started varying n and m within the range [−2, 2] with a difference of 0.1 between two consecutive values tried. The initial range of values for (n, m) was selected taking into account the values needed to fit the kinetic models from the literature, which are provided in Table 1. Thus, the initial possible values for this couple of parameters (n, m) were (−2, −2), (−2, −1.9), (−2, −1.8), (−2, −1.7) … (−1.9, −2), (−1.9, −1.9), (−1.9, −1.8), (−1.9, −1.7) … (2, −2), (2, −1.9), (2, −1.8), (2, −1.7) … For each couple, we calculated the value of ($R^{2}$), when conducting the linear fitting to g versus t, g being calculated through Equation (7). The couple that yielded the value of $R^{2}$ closest to 1 was used as the starting point for the next step of the optimization process. Let us suppose those values were (−1.5, 1). We repeated the procedure with n and m varying within the range [−1.6, −1.4] and [0.9, 1.1], respectively. In this second step, we decreased to 0.01 for the difference between two consecutive values. Therefore, the possible values were (−1.6, 0.9), (−1.6, 0.91), (−1.6, 0.92), (−1.6, 0.93) … (−1.59, 0.9), (−1.59, 0.91), (−1.59, 0.92), (−1.59, 0.93) … (−1.4, 0.9), (−1.4, 0.91), (−1.4, 0.92), (−1.4, 0.93) … Again, the couple that resulted in the value of $R^{2}$ closest to 1 was used as the starting point for the next step of the optimization process and so on. The process might be repeated indefinitely. In this work, we determined the values of *n* and *m* with a resolution of three decimal places.

Since it is fulfilled that:(9)$$\ln k=\ln A-\frac{E}{RT}$$

Using a set of isothermal curves, the pre-exponential factor and the apparent activation energy can be determined from the slope of the line lnk versus 1/T. Thus, according to Equation (9), the slope, S, of the best-fitting line to lnk versus 1/T and the apparent activation energy are related by E=−R·S, while the relationship between the intercept, I, and the pre-exponential factor is A=expI.

### 3.1. Simulation Using the Tridimensional Avrami–Erofeev Equation A3

A set of isothermal curves (Figure 1a) was simulated using the Runge–Kutta method by assuming isothermal conditions, i.e., 600, 610, 620, and 630 K, an activation energy, *E* = 150 kJ·mol^−1^, a preexponential factor, *A* = $10^{10}$ s^−1^, and a tridimensional Avrami–Erofeev model given by:(10)$$f\left(\alpha \right)=3\left(1-\alpha \right){\left[-\ln\left(1-\alpha \right)\right]}^{2/3}$$

When simulated curves are fitted by kinetic models different from the one used in the simulations, the plot of g(α) versus time does not show a linear trend. As a way of example, Figure 1b depicts the plot of g(α) versus time for an A2 kinetic model. As clearly shown in the figure, the dots do not follow the linear trend as the kinetic model is not the correct one. Nevertheless, data are linearized by employing the MSBE equation with n = 0.748 and m = 0.693 (Figure 1c). These n and m values were determined by employing the optimization procedure described above. Figure 1d shows some ideal kinetic models from the literature normalized to their value for α=0.5 together with the MSBE with the *n* and *m* values resulting from the optimization procedure. As expected, the MSBE with n = 0.748 and m = 0.693 perfectly matches the ideal kinetic model A3 that was used for the simulation, yielding straight lines in Figure 1c. From the slope of such lines, values of the rate constant, *k*, for different temperatures are calculated (Equation (3)). Figure 2 shows lnk as a function of 1/T for the data plotted in Figure 1c. According to Equation (9), the pre-exponential factor and the apparent activation energy can be calculated from the intercept and slope, respectively, of the best-fitting line in Figure 2, yielding E=150 kJ and $A=10^{11}$ s^−1^. These values are identical to those used to simulate the curves represented in Figure 1a.

The method presented here can be applied to the most commonly used ideal kinetic models from the literature, often employed for analysing gas–solid reactions. Table 1 collects the values of the parameters n and m needed to reconstruct some ideal models [34]. These values were determined by fitting Equation (5) to the f(α)  ideal model equation.

### 3.2. Simulation Using the Contracting Volume Model R3 with Particle Size Distribution

For a solid material consisting of spheres with a lognormal particle size distribution, PSD, (Figure 3 with standard deviation σ=0.5  and expected value μ= ln(10^−5^)) that thermally decomposes according to the contracting volume kinetic model (R3), it has been previously demonstrated that the extent of reaction of particles with radius r can be expressed as a function of time by [31]:(11)$$\alpha_{r}\left(t\right)=1-{\left(1-\frac{k^{\prime }}{r}t\right)}^{3}$$

$k^{\prime }=A’\cdot \exp\left(-E/RT\right)$ is the inward growth velocity with A’=constant. Note that, when $k^{\prime }t=r$, the reaction is completed. Thus, the overall value of the extent of the reaction considering the PSD is given by:(12)$$\alpha \left(t\right)=\sum_{R}\alpha_{r}\left(t\right)\frac{1}{r\sigma \sqrt{2\pi }}\exp\left(-\frac{{\left(\ln r-\mu \right)}^{2}}{2\sigma^{2}}\right)\Delta R$$
where Δr is the interval of sizes in which the volume fraction is considered to be constant.

Using Equations (4) and (5) and assuming isothermal conditions (600, 610, 620, and 630 K), four curves with E=150 kJ·mol^−1^ and $A^{\prime }=$3.36·10^4^ m·s^−1^ were simulated (Figure 4a). Table 2 collects the values of the *R*-squared correlation coefficient ($R^{2}$) obtained when using the integral form of some ideal kinetic models to determine which one best linearizes g(α) as a function of time.

These ideal models fail at yielding a linear relationship; not even the 3D diffusion model (D3), which has the highest correlation coefficient, provides a linear relationship for g(α) vs. t, as clearly shown in Figure 4b. On the contrary, the MSBE with n = 1.584 and m = −0.019 linearizes the data with $R^{2}=0.9999$, as seen in Figure 4c where data calculated from the simulation are plotted as dots and the best-fitting lines are represented as dashed lines. The values of n and m were determined using the optimization procedure described above.

Figure 4d shows the ideal kinetic models and the MSBE (n = 1.584 and m = −0.019) normalized to their values when α= 0.5 as a function of the extent of the reaction. The normalized values of the derivative, calculated as $\left(\Delta \alpha /\Delta t\right)/\left(\Delta \alpha /\Delta t\right){|}_{\alpha =0.5}$ from the data shown in Figure 4a, were also plotted in the same graph as the open circles. As expected from the results discussed before, the normalized derivative matches the MSBE with n = 1.584 and m = −0.019.

Figure 5a shows lnk as a function of 1/T for the curves plotted in Figure 4b (assuming a D3 kinetic model) and Figure 4c (MSBE obtained from the optimization procedure). As observed, the points follow a linear trend, the best-fitting lines being parallel to each other, which indicates that the value of the activation energy, E = 150 kJ·mol^−1^, is identical for both models (D3 and MSBE). This fact agrees with our previous work where it was mathematically proven that the apparent activation energy values can be determined regardless of the model used to fit the experimental data recorded under isothermal conditions [26]. Actually, this is a unique feature of isothermal kinetic analysis. Nonetheless, the pre-exponential factor does depend on the considered model: A = 3.13 · 10^11^ s^−1^ for the MSBE and A = 5.60 · 10^10^ s^−1^ for D3.

Two sets of isothermal curves (T = 600 K) were simulated using the Runge–Kutta method and the kinetic parameters obtained, as explained above, for the D3 (Figure 5b) and the MSBE (Figure 5c) models and were plotted together with the simulated curve obtained for an R3 kinetic model with a particle size distribution (dots) used for the analysis. It is quite clear that only the curve for the MSBE model matches the curve simulated with a particle size distribution. These results are consistent with previous works that highlight the importance of selecting the proper kinetic model to reconstruct the experimental data [30]. Actually, with the parameter values obtained for the kinetic model D3, the time predicted to achieve α=0.8 is 156 s, which is ~50% larger than the actual time interval required to reach that value of the extent of the reaction. On the other hand, the curve built using the model based on the MSBE perfectly matches the simulated curve.

A common approach to validate the results of the isothermal kinetic analysis consists of using the parameters and the model obtained to predict the results of experiments conducted under conditions different from isothermal ones. Figure 6 shows the results of simulating a linear heating experiment with a heating rate of 5 °C/min and the same kinetic parameters considered for the simulation of the isothermal curves. The predictions using the two models compared in Figure 5 are plotted as continuous lines. The perfect agreement between the data simulated and the MSBE (n = 1.584 and m=−0.019) is clear, whilst the prediction using the kinetic model D3 fails at fitting the curve.

## 4. Application to an Experimental Case: The Thermal Decomposition of Ethylene-Propylene-Diene

Figure 7a includes the time evolution of the extent of the reaction during the isothermal pyrolysis of EPDM under conditions described in the experimental section. As shown in Figure 7b, the data can be linearized by employing the MSBE equation with n = 1.054 and m = 0.455. The obtained kinetic model resembles a scission kinetic model, as might be observed in Figure 7c, where two scission models have been also plotted for comparison with the resulting MSBE. Indeed, scission has been previously identified as the kinetic mechanism driving the thermal decomposition of EPDM and other polymers [39,40]. Furthermore, the values of the apparent activation energy and the pre-exponential factor determined from the Arrhenius plot (Figure 7d), E = 242 kJ·mol^−1^ and A = 2.4·10^17^ min^−1^, respectively, are in good agreement with those previously reported for this material [40].

To test our capability to predict the results of experiments conducted in conditions different from isothermal, using the kinetic parameters and the model obtained, we conducted experiments on the pyrolysis of EPDM under linear heating conditions employing two different heating rates, namely, 2.5 K/min and 10 K/min. The results of these experiments are plotted as dots in Figure 8, while the predictions obtained using the kinetic parameters of the analysis are represented as solid lines. As might be observed, there is a good agreement between the prediction and the experimental results, which validates the kinetic analysis of EPDM’s pyrolysis.

## 5. Conclusions

In this work, we presented a novel method for the kinetic analysis of isothermal integral data that, unlike conventional analyses, allows for the determination of the kinetic parameters and the model without previous assumptions. The procedure relies on a modified version of the Sestak–Berggren equation to find the kinetic model via the numerical integration of experimental data and optimization.

It was demonstrated that this method can be successfully applied to reactions obeying the ideal kinetic models from the literature. In particular, the procedure was applied to simulated data generated assuming the Avrami–Erofeev kinetic model A3, allowing for the equivalent Sestak–Berggren equation and the kinetic parameters to be obtained with accuracy.

Furthermore, the procedure covers deviations produced by nonideal conditions unconsidered when deriving the kinetic models, such as nonuniform particle size distributions. In this work, this was demonstrated in the case of a contracting volume kinetic model modified by a lognormal particle size distribution.

Finally, the method was validated by its application to experimental data corresponding to the thermal decomposition of EPDM.

The analysis tool presented in this work might be employed for studying any process obeying a unique kinetic model with a single value of the apparent activation energy. Nonetheless, further research needs to be conducted to derive a similar tool for analysing reactions happening in two or more steps with different kinetic models and values of the activation energy.

## Figures and Tables

**Figure 1 materials-16-01851-f001:**
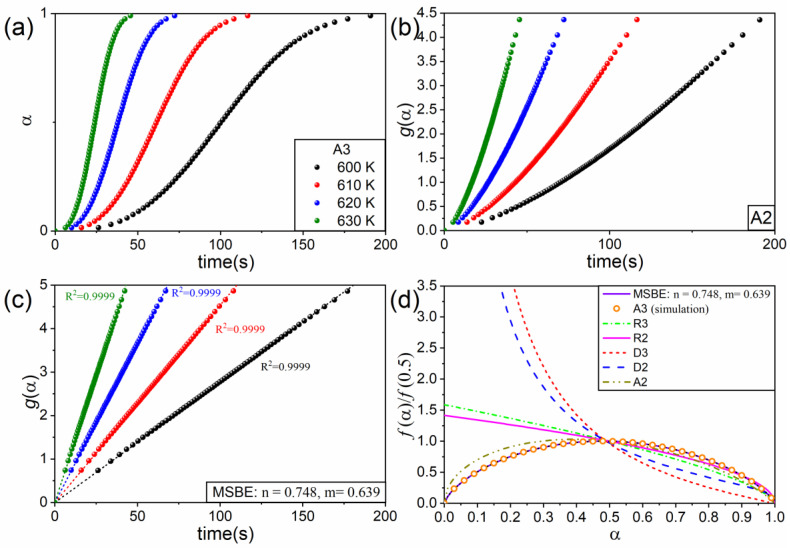
(**a**) Data simulated under isothermal conditions (600, 610, 620, and 630 K) using the kinetic model A3 with E=150 kJ·mol^−1^ and $A=10^{11}$ s^−1^. (**b**,**c**) Plot of g(α) vs. t for the ideal kinetic model A3 and the MSBE with n = 0.748 and m = −0.693. (**d**) Normalized kinetic models used in this study.

**Figure 2 materials-16-01851-f002:**
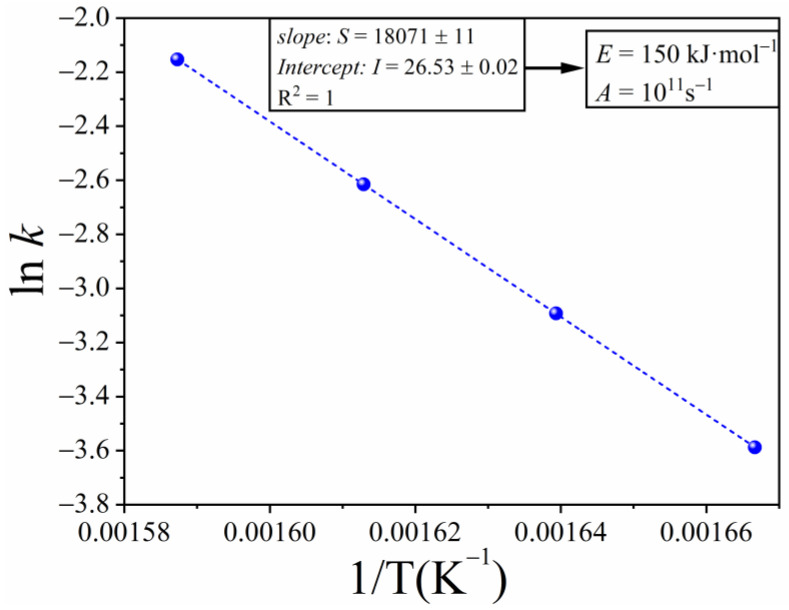
lnk as a function of 1/T. The values of the constant rate are equal to the slope of the best-fitting line of the data plotted in Figure 1c.

**Figure 3 materials-16-01851-f003:**
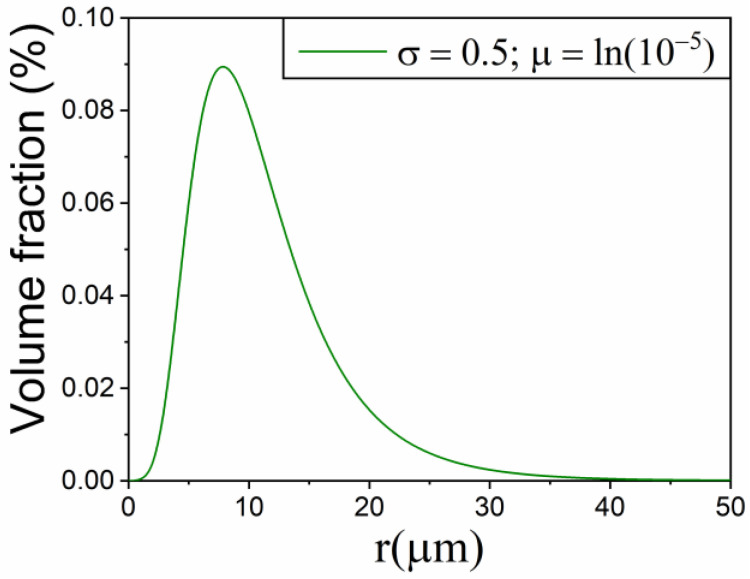
lognormal particle size distribution with σ=0.5 and μ= ln(10^−5^).

**Figure 4 materials-16-01851-f004:**
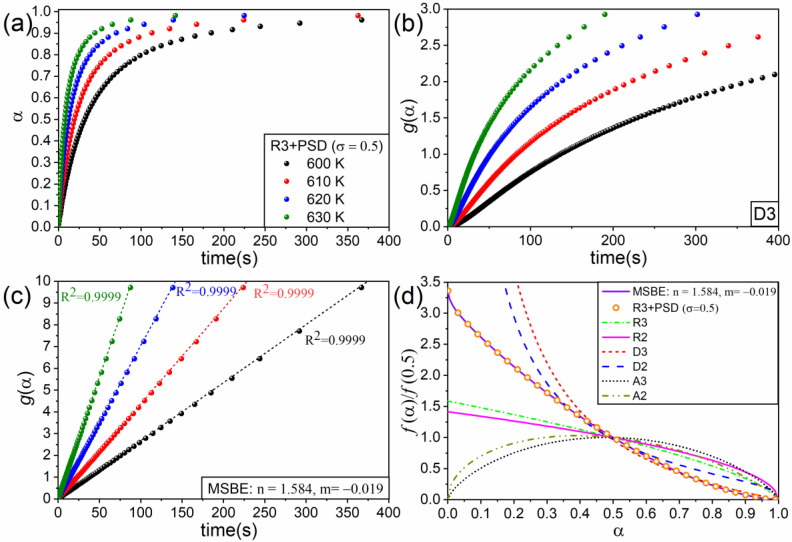
(**a**) Data simulated under isothermal conditions (600, 610, 620, and 630 K) using Equations (4) and (5) and assuming the PSD represented in Figure 3. Subfigures (**b**,**c**) respectively show g(α) vs. t for the ideal kinetic model D3 and the MSBE with n = 1.584 and m = −0.019. (**d**) Normalized kinetic models used in this study.

**Figure 5 materials-16-01851-f005:**
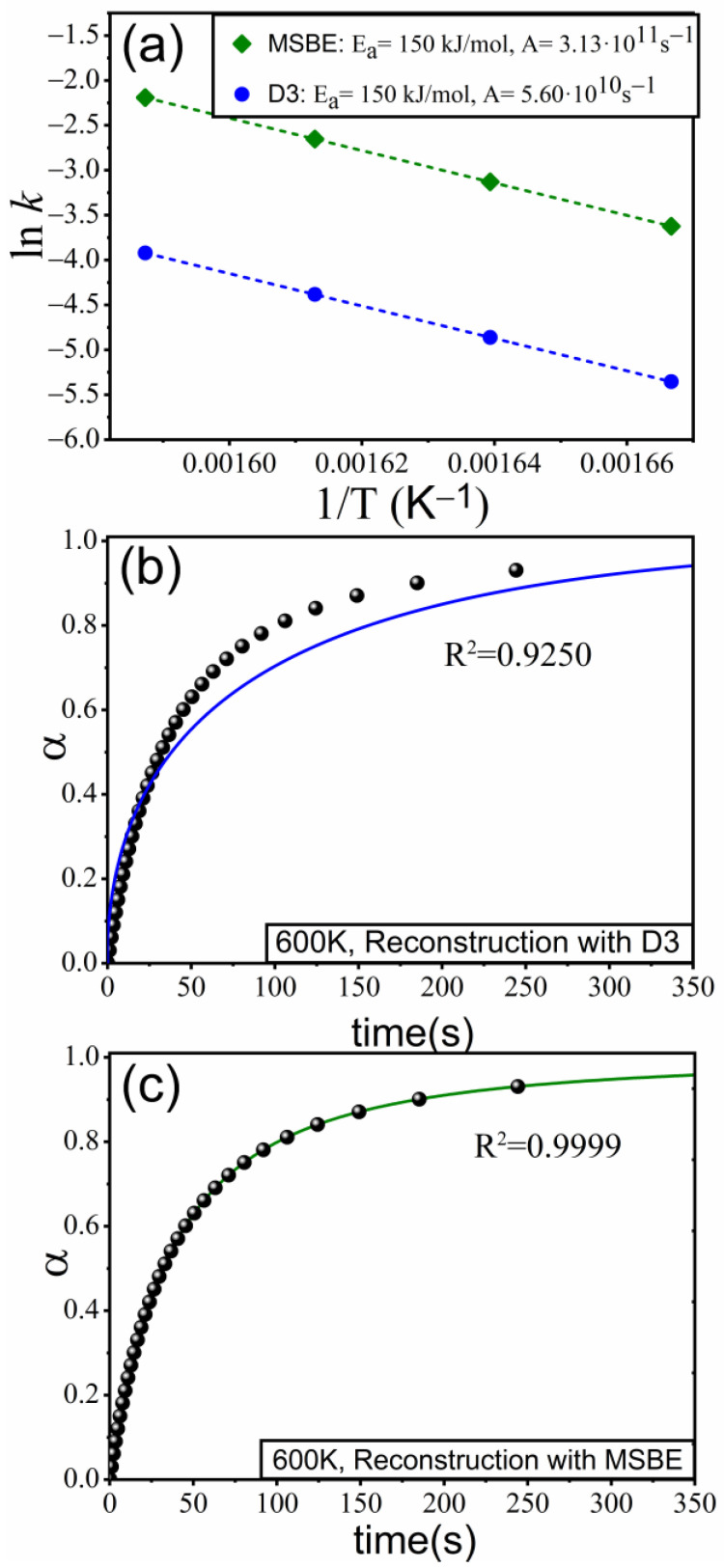
(**a**) Arrhenius plot: lnk as a function of 1/T for the kinetic models D3 and the MSBE (n = 1.584 and m = −0.019). (**b**,**c**) Comparison between the curve simulated at T=600 K and the curves predicted with the parameters obtained with the models D3 and the MSBE, respectively.

**Figure 6 materials-16-01851-f006:**
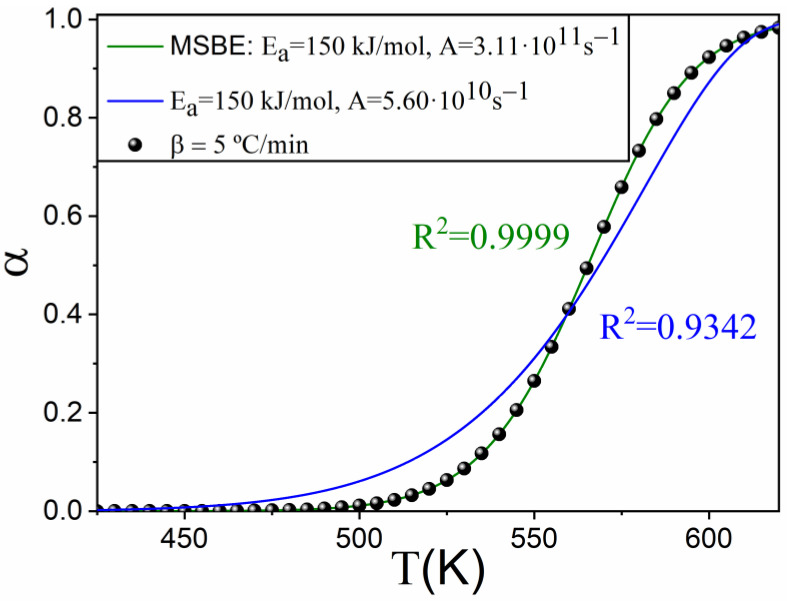
Comparison between the curve simulated under a linear heating rate of 5 K/min (dots) and the predicted data using the kinetic parameters obtained in the isothermal kinetic analysis by assuming a D3 kinetic model (blue solid line) or using the MSBE procedure.

**Figure 7 materials-16-01851-f007:**
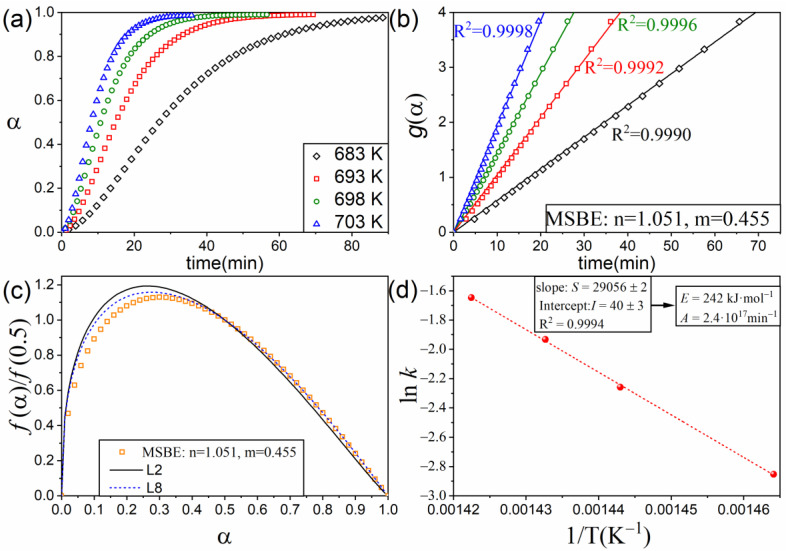
(**a**) Experimental data corresponding to the pyrolysis of EPDM in isothermal conditions: 683, 693, 698, and 703 K. (**b**) Plot of g(α) vs. t for the MSBE with n = 1.051 and m = 0.455. (**c**) Arrhenius plot: lnk as a function of 1/T. (**d**) Comparison between the MSBE obtained in the analysis and the scission kinetic models L2 and L8.

**Figure 8 materials-16-01851-f008:**
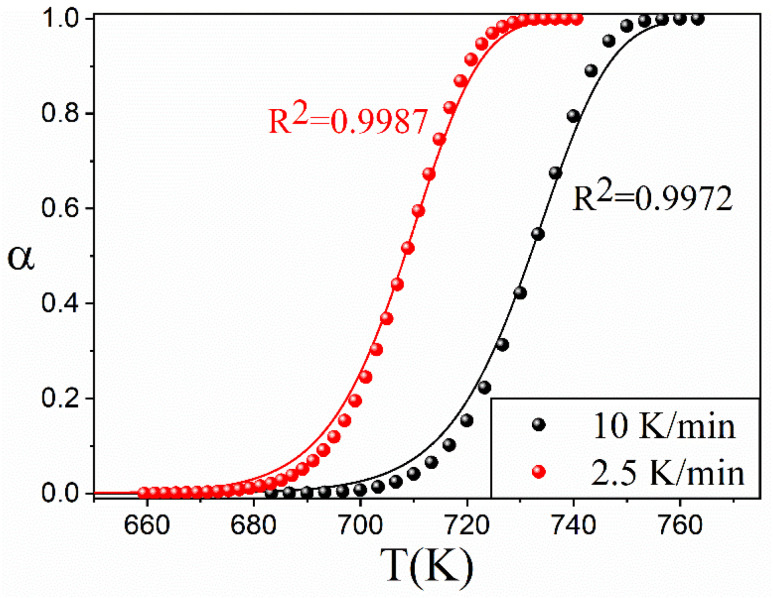
Comparison between the curves recorded under linear heating conditions (2.5 K/min and 10 K/min) and the data predicted using the kinetic parameters obtained in the isothermal kinetic analysis. Predictions are plotted as solid lines, whilst experimental data are represented by dots.

**Table 1 materials-16-01851-t001:** Values of the parameters *n* and *m* needed for reconstructing some kinetic models from the literature.

Kinetic Model	f(α)	n	*m*
Contracting area: R2	${\left(1-\alpha \right)}^{1/2}$	1/2	0
Contracting volume: R3	$${\left(1-\alpha \right)}^{2/3}$$	2/3	0
1D Avrami–Erofeev equation: F1	(1−α)	1	0
2D Avrami–Erofeev equation: A2	$$2\left(1-\alpha \right){\left[-\ln\left(1-\alpha \right)\right]}^{1/2}$$	0.806	0.515
3D Avrami–Erofeev equation: A3	$$3\left(1-\alpha \right){\left[-\ln\left(1-\alpha \right)\right]}^{2/3}$$	0.748	0.693
2D diffusion: D2	1/[−ln(1−α)]	0.425	−1.008
3D diffusion, Jander equation: D3	$$[3{\left(1-\alpha \right)}^{2/3}]/\left\{2\left[1-{\left(1-\alpha \right)}^{1/3}\right]\right\}$$	0.951	−1.004

**Table 2 materials-16-01851-t002:** Values of $R^{2}$ obtained in the linear fitting of g(α) versus time using different ideal kinetic models from the literature and the MSBE model obtained by optimization.

Kinetic Model	$\langle R^{2}\rangle$
MSBE: n = 1.584, m = −0.019	0.9999
R3	0.6844
R2	0.6283
A2	0.5823
A3	0.4934
D2	0.7642
D3	0.9893

## Data Availability

The data will be made available on request from the corresponding author.

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
