# Peer review of "Flexible Kinetic Model Determination of Reactions in Materials under Isothermal Conditions"

_materials, 2023, doi:10.3390/ma16051851_

Round 1

Reviewer 1 Report

The work presents a novel method for kinetic analysis of isothermal reaction data. Without assuming the kinetic model, this method uses a general kinetic equation to find the functional form of the kinetic model via numerical integration and optimization. pre-exponential factor and apparent activation energy can be accurately fitted through the procedure. The theoretical derivation is clear and comprehensive, and the method is fully tested. I just have few issues as listed below:

1.     Experimental section, line 79-84. The authors state that “the experimental conditions were very carefully selected”. Can the author provide detail information on the condition selection rather than just saying “carefully”.

2.     a is the extent of reaction. Can the author provide an example? Like the concentration of the product?

3.     The author introduced MSBE in equation (5). Please provide a brief introduction of the parameters used. Are they just empirical constants?

4.     Page 5, line 162. The author should specify the type of the ideal kinetic model that current method can be applied. Can this method be applied to gas phase kinetic model ?

5.     Page 6, line 170. What is the meaning of s and m.

Reviewer 2 Report

After a detailed reading of the manuscript and overall literature review can be concluded that manuscript could be accepted for the publication after resolving the following issues:

1-The Authors should avoid lumping references, especially in the introduction section. Each referenced source should be briefly elaborated on and put in the context of the performed study.

2-The selection of initial possible values for a couple of parameters (n, m) should be further elaborate and explained.

3-Authors should define the S and I values in Figures 2 and 7.

4-On all Figures where the Authors provide the modeling and experimental results comparison, the fitting quality should be provided.

5-The Conclusions should be extended with the wider findings in order to further improve the novelty presented in this manuscript. Furthermore, the Authors should provide the recommendations and limitations of using the proposed novel methodology.
